# COVA1-18 neutralizing antibody protects against SARS-CoV-2 in three preclinical models

Pauline Maisonnasse [1,21], Yoann Aldon [2,21], Aurélien Marc[3], Romain Marlin [1], Nathalie Dereuddre-Bosquet [1], Natalia A. Kuzmina [4,5], Alec W. Freyn [6], Jonne L. Snitselaar[2], Antonio Gonçalves [3], Tom G. Caniels [2], Judith A. Burger[2], Meliawati Poniman[2], Ilja Bontjer [2], Virginie Chesnais[7], Ségolène Diry[7], Anton Iershov[7], Adam J. Ronk [4,5], Sonia Jangra[6], Raveen Rathnasinghe[6,8], Philip J. M. Brouwer[2], Tom P. L. Bijl[2], Jelle van Schooten [2], Mitch Brinkkemper[2], Hejun Liu[9], Meng Yuan [9], Chad E. Mire [5,10], Mariëlle J. van Breemen[2], Vanessa Contreras[1], Thibaut Naninck [1], Julien Lemaître[1], Nidhal Kahlaoui[1], Francis Relouzat[1], Catherine Chapon[1], Raphaël Ho Tsong Fang[1], Charlene McDanal[11], Mary Osei-Twum[12], Natalie St-Amant[12], Luc Gagnon[12], David C. Montefiori[11], Ian A. Wilson [9], Eric Ginoux[7], Godelieve J. de Bree[13], Adolfo García-Sastre [6,14,15,16], Michael Schotsaert [6,16], Lynda Coughlan [6,17], Alexander Bukreyev [4,5,10], Sylvie van der Werf [18,19], Jérémie Guedj [3], Rogier W. Sanders[2,20 ✉], Marit J. van Gils[2 ✉] & Roger Le Grand [1 ✉]

Effective treatments against Severe Acute Respiratory Syndrome coronavirus 2 (SARS-CoV-2) are urgently needed. Monoclonal antibodies have shown promising results in patients. Here, we evaluate the in vivo prophylactic and therapeutic effect of COVA1-18, a neutralizing antibody highly potent against the B.1.1.7 isolate. In both prophylactic and therapeutic settings, SARS-CoV-2 remains undetectable in the lungs of treated hACE2 mice. Therapeutic treatment also causes a reduction in viral loads in the lungs of Syrian hamsters. When administered at 10 mg kg-1 one day prior to a high dose SARS-CoV-2 challenge in cynomolgus macaques, COVA1-18 shows very strong antiviral activity in the upper respiratory compartments. Using a mathematical model, we estimate that COVA1-18 reduces viral infectivity by more than 95% in these compartments, preventing lymphopenia and extensive lung lesions. Our findings demonstrate that COVA1-18 has a strong antiviral activity in three preclinical models and could be a valuable candidate for further clinical evaluation.

A full list of author affiliations appears at the end of the paper.

Across the world, the Coronavirus Disease 19 (COVID-19) pandemic caused by severe acute respiratory syndrome coronavirus 2 (SARS-CoV-2) continues to escalate[1]. Despite the progressive rollout of vaccines, there remains an urgent need for both curative and preventive measures, especially in individuals with high risk. Monoclonal neutralizing antibodies (NAbs), isolated from convalescent COVID-19 patients, are one of the most promising approaches and two NAb-based products have already received emergency use authorizations by regulatory agencies in both the US[2,3] and Europe[4,5]. Although their clinical efficacy in hospitalized patients remains to be fully assessed, their capability to reduce viral loads and hospitalization in high risk individuals shows that NAbs constitute an effective treatment when administered early enough after symptom onset[6–8].

We and others have previously isolated and characterized several highly potent monoclonal NAbs with half-maximum inhibitory concentration (IC$_{50}$) values in the picomolar range[9–12], with the majority of these targeting the receptor binding domain (RBD) on the S1 subunit of the S protein. We previously identified COVA1-18, an RBD-specific monoclonal Ab, as one of the most potent NAb in vitro[9].

In this work, we use three different experimental models as well as mathematical modeling to demonstrate that COVA1-18 rapid and extensive biodistribution is associated with a very potent antiviral effect, and make it a promising candidate for clinical evaluation, both as a prophylactic or therapeutic treatment of COVID-19.

## Results

**COVA1-18 in vitro potency is dependent on avidity**. To advance our earlier in vitro results[9] on COVA1-18 and allow for better comparability with other studies, we used two pseudovirus assays, one using lentiviral pseudotypes with an ACE2-expressing 293 T cell line[13], and one using VSV-pseudotypes with Vero E6 cells[14], to confirm the potency of COVA1-18. With these assays, we found that COVA1-18 IgG inhibited lentiviral SARS-CoV-2 pseudovirus with an IC$_{50}$ of 1.7 ng ml$^{-1}$ (11.3 pM) and VSV-based pseudovirus with an IC$_{50}$ of 9 ng ml$^{-1}$ (60 pM), confirming the remarkable potency previously observed against authentic virus[9] (Supplementary Fig. 1a, Table 1). These results were corroborated in multiple independent labs and COVA1-18 was also equipotent against the D614G variant (Table 1) that now dominates worldwide[15–19] as well as the recently emerged B.1.1.7 variant that includes the N501Y mutation in the RBD[20,21] (Table 2).

COVA1-18 bound strongly to SARS-CoV-2 S protein and showed no cross-reactivity with S proteins of SARS-CoV, MERS-CoV and common cold coronaviruses HKU1-CoV, 229E-CoV and NL63-CoV (Supplementary Fig. 1b)[9]. Biolayer interferometry experiments showed that COVA1-18 IgG bound to soluble SARS-CoV-2 S protein with an apparent dissociation constant (K$_D$) of 5 nM, and its affinity for RBD was similar at 7 nM (Fig. 1a, Supplementary Fig. 1c, d, Table 1). Its Fab displayed a 12-fold weaker binding to RBD compared to IgG (84 nM), with the difference mainly caused by a faster Fab off-rate (Fig. 1a, Table 1), as also observed in a different assay setting (Supplementary Fig. 1d). With an IC$_{50}$ of 199 ng ml$^{-1}$, the COVA1-18 Fab was 237-fold less potent at neutralizing SARS-CoV-2 pseudovirus, showing that the IgG avidity effect is important for COVA1-18 neutralization potency (Supplementary Fig. 1a, Table 1).

**COVA1-18 inhibits viral replication in rodents**. We sought to evaluate whether COVA1-18 could control SARS-CoV-2 viral infection in a previously described Ad5-hACE2 mouse model[22,23] using a 10 mg kg$^{-1}$ dose. COVA1-18 administered intraperitoneally 24 h either prior to or after a SARS-CoV-2 challenge with 10$^4$ plaque forming units (PFU) ($n = 5$ for treated groups, $n = 3$ for control group) was fully protective with no detectable viral replication in the lungs (Fig. 1b, c). We then tested the efficacy of COVA1-18 in the golden Syrian hamster model ($n = 5$ per group), which is naturally susceptible to SARS-CoV-2 and develops severe pneumonia upon infection[24]. We evaluated the effect on lung viral loads of 10 mg kg$^{-1}$ of COVA1-18 given 24 h after a 10$^5$ PFU intranasal challenge (Fig. 1b, d). At 3 days post-infection (d.p.i.), 3/5 animals had high serum neutralization, while for 2/5 animals, low neutralization activity was observed (Supplementary Fig. 1e). On day 3, the COVA1-18 treated group had significantly lower median lung viral titers compared to the control group (3.5 vs 6.7 log$_{10}$ PFU g$^{-1}$, respectively, $p < 0.01$) with lowest viral titers corresponding to the higher neutralizing serum activity (Fig. 1d). The time of treatment (24 h post-infection) and 3-day study period did not allow for prevention of lung damage and recovery monitoring in this model (Supplementary Fig. 1f, g and Supplementary Table 1).

**COVA1-18 PrEP prevents infection in NHP**. We evaluated the potential of COVA1-18 to prevent SARS-CoV-2 infection in cynomolgus macaques in a pre-exposure prophylaxis (PrEP) study. The animals were treated intravenously 24 h prior to viral challenge with a dose of 10 mg kg$^{-1}$ of COVA1-18 (Fig. 2a). Treated and control animals ($n = 5$ per group) were challenged on day 0 with 10$^6$ PFU of SARS-CoV-2 via combined intranasal and intratracheal routes using an experimental protocol developed previously[25,26]. On the day of challenge, the mean COVA1-18 serum concentration was $109 \pm 2.7$ µg ml$^{-1}$ (Fig. 2b, Supplementary Fig. 2a). COVA-18 was also detected in all respiratory tract samples and rectal samples (Fig. 2c–e, Supplementary Fig. 2b–d), and represented on average 1.5% and 1.2% of the total IgG in heat-inactivated content in the nasopharyngeal and tracheal mucosae, respectively. These levels remained constant throughout the study period and similar levels were detected at 3 d.p.i. in bronchoalveolar lavages (BAL) and saliva (Fig. 2e, f). As SARS-CoV-2 can cause damage to non-respiratory organs, we performed a pharmacokinetic study on two additional macaques to characterize the COVA1-18 distribution within the first 24 h using non heat-inactivated samples (Fig. 2g and Supplementary

**Table 1 BLI and neutralization potency of COVA1-18 IgG vs Fab in HEK293T hACE2 cells.**

| | | IC$_{50}$ | | | | BLI | | | | | |
| | | AMC ($n \geq 4$) | | Duke ($n = 1$) | Duke D614G ($n = 1$) | Nexelis ($n = 1$) | RBD loaded ($n = 3$) | | | Soluble S loaded ($n = 3$) | | |
| | | ng ml$^{-1}$ | pM | ng ml$^{-1}$ | | | K$_D$ (nM) | Ka (M$^{-1}$s$^{-1}$) | Kd (s$^{-1}$) | K$_D$ (nM) | Ka (M$^{-1}$s$^{-1}$) | Kd (s$^{-1}$) |
|---|---|---|---|---|---|---|---|---|---|---|---|---|
| 1–18 | IgG | 1.7 | 11.3 | 9.0 | 7.0 | 9.0 | 7.0 | 1.7E+05 | 1.3E-03 | 5.0 | 3.7E+05 | 1.9E-03 |
| | Fab | 199.0 | 3968.0 | N/A | N/A | N/A | 84.1 | 5.0E+04 | 4.1E-03 | N/A | N/A | N/A |

AMC and Duke neutralization assays use lentiviral pseudotyped particles and HEK293T hACE2 cells. Nexelis neutralization assay uses VSVΔG pseudotyped particles and Vero E6 cells. BLI biolayer interferometry, RBD receptor binding domain.

**Table 2 COVA1-18 and COVA1-16 neutralization potency against variants in HEK293T hACE2 cells.**

| | IC$_{50}$ (ng ml$^{-1}$) | | |
|---|---|---|---|
| | **COVA1-18** | **COVA1-16** | **COVA1-18 + COVA1-16 (1:1)** |
| Wild type | 1.7 | 80.7 | N/A |
| D614G | 0.7 | 109.2 | 1.0 |
| B.1.1.7 | 1.4 | 90.8 | 2.2 |
| E484K | >12500 | 94.3 | N/A |
| B.1.351 | >50000 | 50.6 | 97.6 |

Mean IC$_{50}$ values with $n \geq 3$, except for the cocktail with $n = 2$. N/A, not assessed.

Fig. 2e, f). COVA1-18 was found in all organs studied, including the lungs, at concentrations of 4 to 22 ng mg$^{-1}$ of tissue, except for the brain where concentrations were substantially lower (250 pg mg$^{-1}$ of tissue) (Fig. 2g). Altogether, these data showed that COVA1-18 administered intravenously was rapidly and efficiently distributed to the natural sites of infection as well as to organs affected by COVID-19 pathology.

Following viral challenge, control animals showed similar genomic (g)RNA and subgenomic (sg)RNA levels and kinetics as previously described[25,26] with median peak viral loads (VL) of 6.4 and 6.2 log$_{10}$ copies per ml at 1-2 d.p.i. in the nasopharyngeal and tracheal swabs, respectively (Fig. 3a). Active viral replication, as assessed by sgRNA levels, peaked at 1-2 d.p.i. in nasopharyngeal

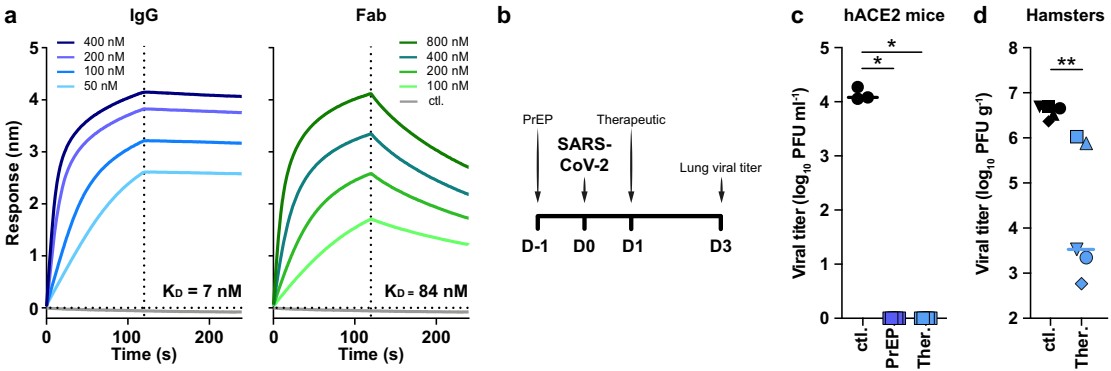

**Fig. 1 COVA1-18 avidity and SARS-CoV-2 protection in rodents. a** Biolayer interferometry sensorgrams comparing COVA1-18 IgG and Fab binding to RBD. K$_D$s are indicated. Representative of 3 independent experiments. **b** Study design with $n = 5$ per group, except mouse control group ($n = 3$). Hamsters were infected with 10$^5$ PFU on day 0 and treated on day 1. Mice received COVA1-18 24 h prior to or after exposure to 10$^4$ PFU. Lung viral titers at 3 days post-infection are shown for mice (**c**) and hamsters (**d**). Bars indicate medians. Mann-Whitney unpaired two-tailed t-test, $p$ values: *:0.0179, **:0.0079. Ctl. control group (black), KD dissociation constant, PFU Plaque forming unit, PrEP pre-exposure prophylaxis (dark blue), Ther. therapeutic (light blue).

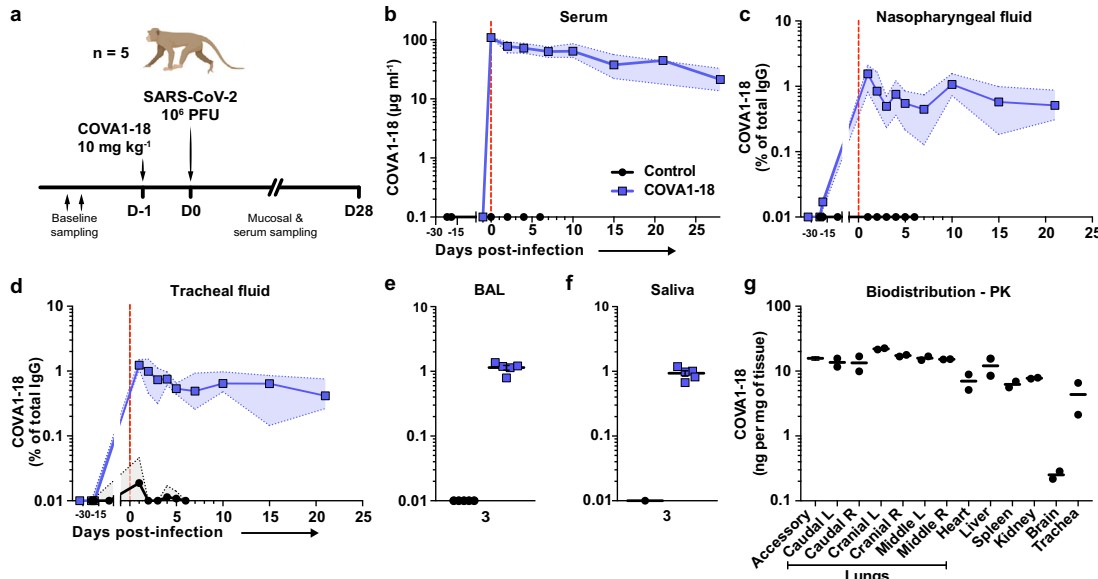

**Fig. 2 COVA1-18 serum and mucosal pharmacokinetic in infected cynomolgus macaques. a** Study design. Two groups of $n = 5$ were exposed to 10$^6$ PFU of SARS-CoV-2 (intranasal and intratracheal routes). Treated animals received 10 mg kg$^{-1}$ of COVA1-18 1 day before challenge. **b** COVA1-18 serum concentration (mean with range). COVA1-18 concentration reported as percent of total cynomolgus IgG in heat-inactivated (**c**) nasopharyngeal fluid, **d** tracheal fluid (means with range), **e** bronchoalveolar lavage (BAL) and **f** saliva (means ± SEMs) with $n = 5$, except for the control group in (**f**) where $n = 1$. **g** The two macaques from the pharmacokinetic study were euthanized at 24 h post-treatment and their organs analyzed to assess the biodistribution of COVA1-18. The concentration of COVA1-18 was normalized to the weight of each sample for every organ. Bars represent means. The red dashed line indicates challenge day. L left, PFU Plaque forming unit, PK pharmacokinetic, R right.

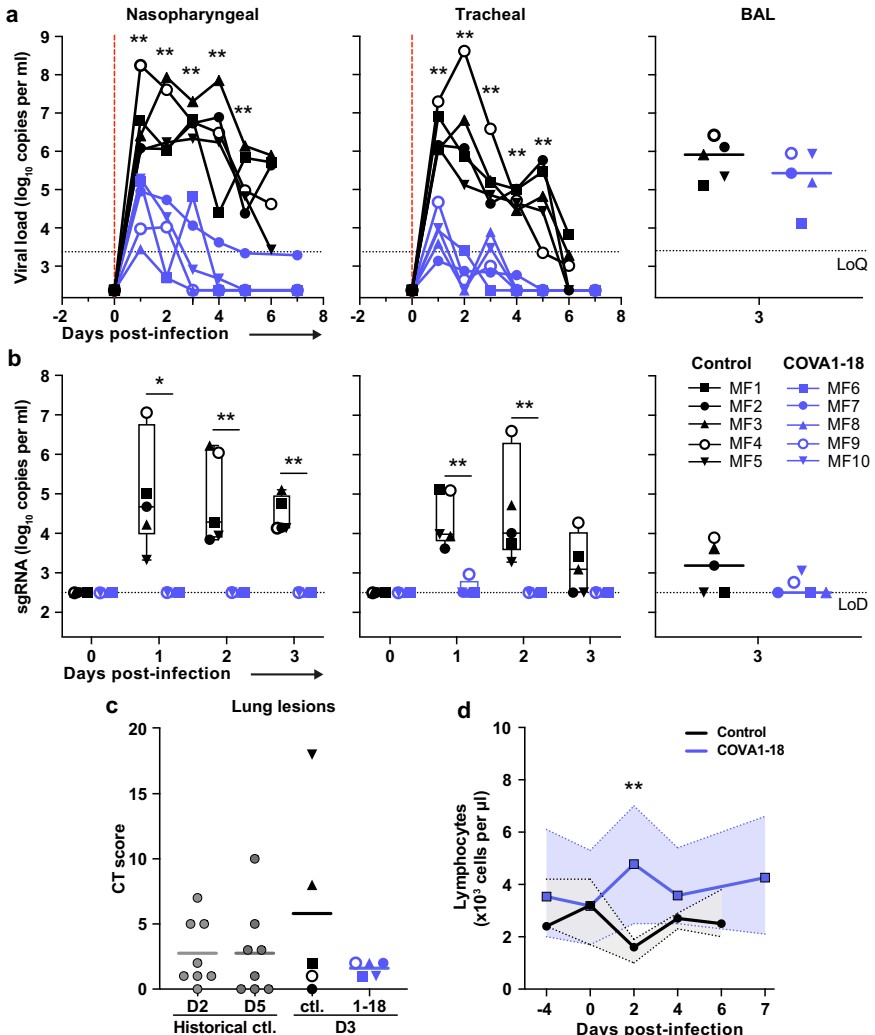

**Fig. 3 COVA1-18 pre-exposure prophylaxis protects cynomolgus monkeys against SARS-CoV-2 challenge and clinical symptoms. a** Genomic (g)RNA and **b** subgenomic (sg)RNA loads determined by PCR in nasopharyngeal fluids (left), tracheal fluids (middle) and bronchoalveolar lavages (BAL) (right). Individual values are plotted for nasopharyngeal and tracheal samples and bars represent medians for BAL. For **b**, boxes and whiskers representation with min-max., median, 25th–75th percentile for n = 5 per group. **c** Chest CT scores were determined at 3 d.p.i. and at 2 or 5 d.p.i for historical controls (n = 8). **d** Absolute lymphocyte count in the blood (mean with range). Mann-Whitney unpaired two-tailed t-test, p values: * < 0.05, ** < 0.01. 1–18, COVA1-18; CT Computed Tomography, Ctl. control group, LoD limit of detection, LoQ limit of quantification.

and tracheal swabs with median values of 4.6 and 4.0 log$_{10}$ copies per ml, respectively (Fig. 3b). At 3 d.p.i., viral loads were detected in the BAL with a median value of 4.9 log$_{10}$ copies per ml of gRNA and 3.2 log$_{10}$ copies per ml of sgRNA, including 3 animals with no detectable sgRNA.

In comparison, treated animals had a reduction of 2.2 and 3.4 log$_{10}$ median gRNA VL in tracheal swabs on days 1 and 2 (both p < 0.01 to controls), and had undetectable VL after day 4 (Fig. 3a). The difference was also evident in nasopharyngeal swabs, with treated animals having a reduction of 1.5 and 2.2 log$_{10}$ gRNA VL on days 1 and 2 (both p < 0.01 to controls). By day 4, 4/5 treated animals had undetectable gRNA in the nasopharyngeal swabs while one animal (MF7) remained positive with a low residual gRNA signal up to 7 d.p.i. COVA1-18 treatment dramatically hindered viral replication in the upper respiratory tract as evidenced by the absence of detectable sgRNA in the nasopharyngeal and tracheal swabs for all treated animals with the exception of animal (MF9) that showed a low signal at 1 d.p.i. only in the tracheal swabs (Fig. 3b). Therefore, in the treated group, most upper respiratory tract gRNA VL likely represents

the progressive elimination of the challenge inoculum, and does not result from active replication. The gRNA and sgRNA loads in BAL were also lower in COVA1-18 recipients compared to controls but the difference did not reach statistical significance (Fig. 3a, b). Cynomolgus anti-S IgM was detected as early as 6 d.p.i. in control animals, while no IgM was detected in treated animals at early timepoints (Supplementary Fig. 2g). Some IgM was detected at 28 d.p.i. in 3 treated animals (MF6, MF7, MF9), although levels remained lower than controls at 6 d.p.i. No anti-S specific cynomolgus IgG was detected up to the day of euthanasia in control animals (7 d.p.i.) or in treated animals up to 28 d.p.i. (Supplementary Fig. 2h). Overall, these results demonstrate that a 10 mg kg$^{-1}$ dose of COVA1-18 PrEP dramatically reduced the acquisition and/or early spread of SARS-CoV-2 in the different respiratory compartments.

Analysis of lung lesions by chest computed tomography (CT) showed that all treated animals had few and small lung lesions as recorded by low CT scores at 3 d.p.i. while 2/5 controls showed mild pulmonary lesions characterized by non-extended ground-glass opacities (GGOs) with scores superior to 5, consistent with

what was observed in historic controls[25] and mirroring the heterogeneity of COVID-19 infection in humans[27] (Fig. 3c). In addition, we observed that all control animals were lymphopenic at 2 d.p.i., consistent with previous studies[25,26], while all treated animals had normal lymphocyte counts throughout the study ($p < 0.01$ for the comparison) (Fig. 3d and Supplementary Fig. 2i).

One concern about SARS-CoV-2 vaccines and NAb treatments is the possible generation of suboptimal concentrations of NAb in individuals, which could foster viral escape[28]. Sequencing analysis of nasopharyngeal, tracheal and BAL samples at 3 d.p.i. showed that COVA1-18 treatment resulted in enrichment of subclonal variations in N and ORF1ab. One mutation (E725G) was detected in the S gene in the MF7 BAL sample when applying standard quality filters, but this mutation has not been previously implicated in immune escape and located outside the epitope of COVA1-18 (Supplementary Fig. 3 and Supplementary Information). The high efficacy of COVA1-18 treatment prevented recovery of viral genetic information past 3 d.p.i.

**Prediction models refine COVA1-18 dosage.** Next, we used a viral dynamic model previously developed in the same SARS-CoV-2 NHP experimental model[29] to evaluate the level of protection conferred by COVA1-18, and guide potential subsequent studies on SARS-CoV-2 MAbs. The model considers a target cell limited infection in both nasopharyngeal and tracheal compartments. In addition to the previously developed model, we assumed that sgRNA was a proxy for the total number of non-productively and productively infected cells (see supplementary methods) and we further assumed that COVA1-18 plasma drug concentrations over time, noted $C(t)$, was the driver of drug efficacy. We modeled the changes in $C(t)$ using a standard first order absorption and elimination model, and we estimated the half-life of COVA1-18 in plasma to be 12.6 days (Supplementary Fig. 4a). We assumed that COVA1-18 reduces infectivity rate in both tracheal and nasopharyngeal compartments with an efficacy, noted $\eta(t)$, determined by the following model $\eta(t) = \frac{C(t)}{C(t)+EC_{50}}$, where $EC_{50}$ is the plasma COVA1-18 concentrations corresponding to a 50% reduction of viral infectivity. The model fitted the viral kinetics well in all animals (Fig. 4a, Supplementary Fig. 4b, Supplementary Table 2). The $EC_{50}$ was estimated to be 2.2 and 0.053 $\mu g \, ml^{-1}$ in the nasopharynx and trachea, respectively, which is roughly 50 and 2000 times lower than the plasma drug concentrations of 109 $\mu g \, ml^{-1}$ observed on the day of infection (see above). Thus, these results confirm that the efficacy of COVA1-18 was very high, with efficacies above 95% and 99.9% in nasopharyngeal and tracheal compartments on the day of infection, respectively (Fig. 4a, Supplementary Fig. 4b). Given the long half-life of the drug, this efficacy was maintained over time, and we estimated that the mean individual efficacy of the COVA1-18 in the first 10 days following infection ranged between 96.67% and 97.50% in the nasopharynx and between 99.91% and 99.94% in the trachea (Supplementary Fig. 4c).

Next, we used our model to investigate changes in experimental conditions, such as COVA1-18 dose being administered at a lower dose and/or after the viral challenge (see methods). In all scenarios considered, a dose of 5 $mg \, kg^{-1}$ was determined to provide nearly similar results than 10 $mg \, kg^{-1}$ (Fig. 4b, c, Supplementary Fig. 5a, b). A dose of 1 $mg \, kg^{-1}$ could be sufficient to prevent active viral replication as long as treatment is given prior to infection, but might be insufficient in a therapeutic setting. However, this dose could be relevant if lower doses of virus were used for infection, such as $10^4$ or $10^5$ PFU (Supplementary Fig. 4d–g).

**COVA1-18/1-16 cocktail neutralizes B.1.351.** Many highly potent RBD-targeting mAbs are affected by mutations in emerging variants-of-concern (VOC), in particular E484K[30,31]. We evaluated the ability of COVA1-18 to neutralize VOCs B.1.1.7 and B.1.351 as well as a E484K single mutant virus. While COVA1-18 retains its high potency against the B.1.1.7 strain, it lost its capacity to neutralize the B.1.351 strain due primarily to the RBD E484K mutation in the spike (Supplementary Fig. 6). Therefore, we also evaluated the in vitro potency of COVA1-18 in a cocktail with COVA1-16, an antibody that neutralizes B.1.351 as well as SARS-CoV-1, but is less potent than COVA1-18[9] (Supplementary Fig. 6 and Table 2). This mAb cocktail retained the high potency of COVA1-18 against wild-type, D614G and B.1.1.7 and also efficiently neutralized B.1.351, providing an avenue for broad mAb prophylaxis and treatment against VOCs.

## Discussion

Despite the recent approval of several SARS-CoV-2 vaccines by health authorities, the slow roll-out of vaccination campaigns will not result in resolution of the pandemic in the immediate future. Furthermore, the emergence of viral escape mutants may lead to reduced vaccine efficacy, and some individuals, such as immunocompromised patients or the elderly, may not mount adequate protective immune responses to vaccination. Thus, there is an urgent need to develop effective therapeutics, in particular for individuals with high risk of severe disease.

In hACE2-expressing mice and golden Syrian hamsters, COVA1-18 showed remarkable control of SARS-CoV-2 infection. These promising results were confirmed in NHPs, with COVA1-18 given one day prior to infection achieving nearly complete protection in the upper respiratory tract in cynomolgus macaques. Using a viral dynamic model, we estimated that COVA1-18 reduced viral infectivity by >95% and 99.9% in nasopharyngeal and tracheal compartments, respectively. The robustness of these results are reinforced by the high challenge dose that we used, which was 10 to 100-fold higher than in other NHP studies evaluating NAbs for PrEP against SARS-CoV-2[32–38]. In fact, the model allowed us to predict, without using additional animals, that a high level of protection could be achieved with lower doses of 5 $mg \, kg^{-1}$ and 1 $mg \, kg^{-1}$ with lower inoculum doses of $10^5$ or $10^4$ PFU, both in prophylactic and therapeutic settings (Supplementary Fig. 4, Supplementary Fig. 5).

How do these levels of efficacy greater than 95% translate into clinical efficacy? In previous work, we estimated that achieving 90% efficacy would be sufficient to confer a high level of protection against infection acquisition if treatment can be administered prophylactically or just after a high-risk contact[38]. In hospitalized patients, where viral load kinetics after admission are associated with the risk of death, we estimated that administration of treatment with an efficacy higher than 90% could reduce the time to viral clearance by more than 3 days in patients over 65 years of age, which could translate into significantly lower rates of mortality in this population[39].

Several NAbs are being developed and some have achieved clinical endpoints, such as the reduction of the risk of hospitalization in patients that initiate treatment within 5 days of symptom onset[6–8], leading to their approval for emergency use[22,32–37,40]. However, the narrow efficacy range of FDA-approved NAbs[41–43], together with rapidly spreading new variants complicate treatment strategies[30,31,44,45], highlighting the need for additional treatment options, including potent NAbs, such as COVA1-18, that could be used in combination with other NAbs. The plasma half-life was 12.6 days, albeit lower to what is found typically for human NAbs in humans[37], ranging from 15 to 25 days, and consistent with values reported for other human NAbs in the macaque model (Supplementary Table 3). The efficacy in this model was high, despite the high challenge dose ($10^6$ PFU) used here. We estimated that with

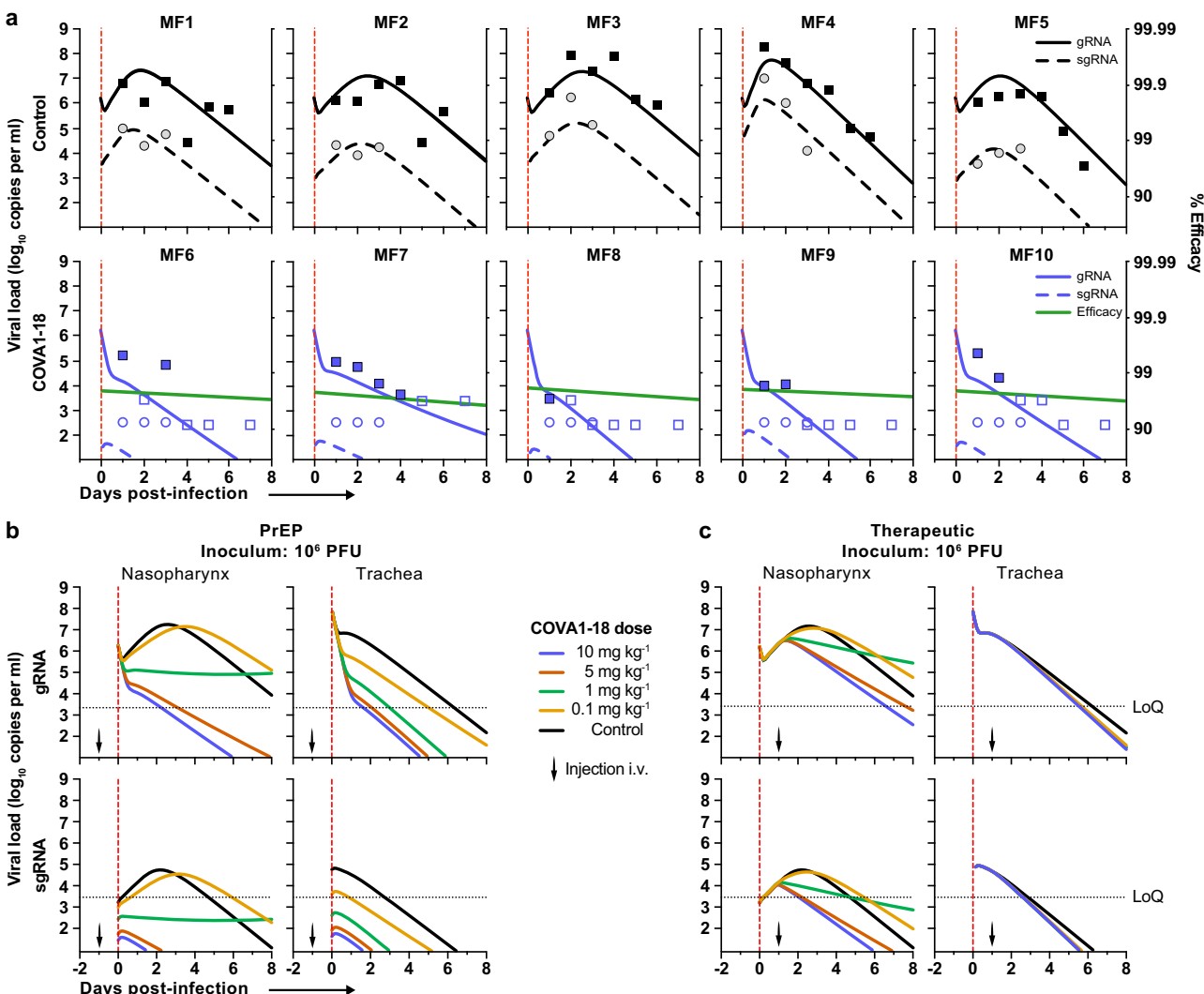

**Fig. 4 Modeling of viral dynamics and treatment efficacy. a** Individual prediction of the nasopharyngeal genomic (g)RNA and subgenomic (sg)RNA in control (top) and treated animals (bottom) with individual efficacy prediction indicated (green line). The dashed red line indicates the time of infection. gRNA (squares) and sgRNA (circles) data are indicated as plain (above LoQ) or open (below LoQ). **b** Model predictions of gRNA and sgRNA dynamics with 4 doses of COVA1-18 given 24 h prior to challenge (arrow). **c** Simulation as in (**b**) with COVA1-18 given 24 h post-infection. Black dotted lines indicate LoQ (limit of quantification), i.v. intravenous, PFU plaque forming units, PrEP Pre-exposure prophylaxis.

lower inoculum doses of $10^4$ or $10^5$ PFU, as used in other studies[32,33,37] (Supplementary Table 3), a dose of 10 mg kg⁻¹ COVA1-18 could reduce the viral load even more dramatically (Supplementary Fig. 4d and Supplementary Fig. 5a, b). Although it is difficult to compare results obtained with different experimental and virological models, this model shows that the in vivo efficacy of COVA1-18 is comparable with what has been obtained for other advanced NAbs in clinical development.

An optimal cocktail should not only be based on intrinsic efficacy against wild-type virus of each NAbs, but rather whether synergy could be achieved in terms of binding domain and/or spectrum of efficacy. Indeed, the increasing prevalence of mutant strains has reduced the sensitivity to pre-existing NAbs, including those given in combination[30]. Escape mutations can arise following single NAb treatment as recently demonstrated[37,46] and the one S mutation found in a unique sample from one animal treated with COVA1-18 is not in the epitope of COVA-18. Importantly, we and others have determined that COVA1-18 retains high potency against the B.1.1.7 variant, which includes the N501Y mutation[20,21]. However, COVA1-18 lost its potency against B.1.351 which harbors the E484K mutation that is also

found in the B.1.1.28 lineage, similar to what has been found with first wave convalescent plasma and many NAbs[30,31]. This finding highlights the necessity of using NAbs cocktails targeting distinct epitopes and we propose the use of the SARS-CoV-1 cross-neutralizing antibody COVA1-16, which can effectively neutralize B.1.351, in a 1:1 cocktail with COVA1-18. In addition, the half-life of COVA1-18 can be extended by incorporating the LS or YTE[47] mutations, which can further reduce the protective dose required and reduce the cost of treatment.

While approved SARS-CoV-2 mAbs are given intravenously, other therapeutic mAbs are given intramuscularly or by sub-cutaneous injection[48]. SARS-CoV-2 mAbs could potentially also be administered intranasally or delivered via gene therapy to the airways[49], to provide protection where it is most needed, i.e. the respiratory tract. The biodistribution of COVA1-18 by different routes of administration would also have to be investigated. In addition, we note that COVA1-18 and numerous potent neutralizing Abs isolated to date against SARS-CoV-2 have very low levels of somatic hypermutation. Thus, these antibodies are very close to the germline precursor and unlikely to trigger anti-idiotypic response in patients.

In conclusion, our COVA1-18 in vitro data translated into a powerful protective drug in three preclinical models to prevent SARS-CoV-2 replication. Together with our prediction model, these data showed that COVA1-18 could be used in patients at low doses either to prevent infection or to reduce viral loads in a therapeutic setting, with a potential greater impact in high-risk patients. The high in vivo efficacy of COVA1-18 and its demonstrated potency against the B.1.1.7. isolate also suggests that it is a promising candidate for a NAb cocktail.

## Methods

**IgG, Fab, and soluble viral protein expression.** COVA1-18 was isolated from a participant in the "COVID-19 Specific Antibodies" (COSCA) study as described[9]. The COSCA study was conducted at the Amsterdam University Medical Centre, location AMC, the Netherlands, and approved by the local ethical committee of the AMC (NL 73281.018.20). COVA1-18 IgG was produced in HEK293F suspension cells as previously described[9]. COVA1-18 His-tagged Fab was produced in Expi-CHO cells as previously described[50]. Spike and RBD proteins were produced and purified as previously described[9]. Briefly, cells were transfected at a density of 0.8–1.2 million cells per mL by addition of a mix of PEImax ($1 \mu g \mu l^{-1}$) with expression plasmids ($312.5 \mu g l^{-1}$) in a 3:1 ratio in OptiMEM. Supernatants of glycoproteins were harvested six days post transfection, centrifuged for 30 min at 4000 rpm and filtered). Constructs with a his-tag were purified by affinity purification using Ni-NTA agarose beads. Protein eluates were concentrated and buffer exchanged to PBS using Vivaspin filters with a 100 kDa molecular weight cutoff (GE Healthcare) for Spike proteins or 10 kDa for RBD. Protein concentrations were determined by the Nanodrop method using the proteins peptidic molecular weight and extinction coefficient as determined by the online ExPASy software (ProtParam).

**Bio-layer interferometry.** The affinity of COVA1-18 IgG and His-tagged Fab versions were determined using Ni-NTA biosensors (ForteBio) onto which $20 \mu g ml^{-1}$ of SARS-CoV-2 RBD was in running buffer (PBS, 0.02% Tween-20, 0.1% BSA) was loaded for 300 s as previously described[50]. The association rate and dissociation step were assessed over a 120 s step each. Serially diluted IgG (50, 100, 200, and 400 nM) and Fab (100, 200, 400, and 800 nM) were tested and an anti-HIV-1 His-tagged Fab at 800 nM in running buffer was included as negative control. $K_D$ were determined using ForteBio Octet CFR software using a 1:2 fitting model for IgGs and a 1:1 fitting model for Fabs. The apparent affinity of COVA1-18 IgG to the SARS-CoV-2 S trimer was determined as described above except that $20 \mu g ml^{-1}$ SARS-CoV-2 S 2 P Fld His protein was loaded instead of RBD. The COVA1-18 IgG avidity effect was further evaluated by titrating the loaded SARS-CoV-2 RBD (5, 1, 0.2, and $0.04 \mu g ml^{-1}$). An additional loading step using His-tagged HIV-1 gp41 was performed to minimize background binding of His-tagged Fabs to the biosensor and both the COVA1-18 IgG and Fab concentrations were set at 250 nM. All other steps were performed as described above. Data were acquired with Octet Data Acquisition 10.0.03.12 and analyzed with Octet Analysis HT 10.0.3.7 (ForteBio).

**Ni-NTA-capture ELISA.** SARS-CoV-2, SARS-CoV, MERS, HKU1, 229E and NL63 S His-tagged proteins were loaded at $2 \mu g ml^{-1}$ in TBS/2% skimmed milk (100 μl/well) on 96-well Ni-NTA plates (Qiagen) for 2 h at room temperature (RT). Three-fold serially diluted COVA NAb were then added onto the plates for 2 h at RT followed by the addition goat anti-human IgG-HRP (Jackson Immunoresearch) secondary Ab (1:3000) for 1 h at RT. The plates were developed for 3 min using TMB solution and then stopped. Optical densities were measured at 450 nm on a spectrophotometer and data graphed using GraphPad Prism software (v8.3.0).

**Detection of human IgG in NHP fluid.** Detection of COVA1-18 in NHP samples determined by ELISA using a protocol adapted from others[33]. Briefly, half area high binding 96-well plates (Greiner Bio-One) were coated overnight with goat anti-Human IgG H+L (monkey pre-adsorbed) at $1 \mu g ml^{-1}$ in PBS. The plates were then blocked in casein buffer (Thermo Scientific) for 2 h at RT. Serum and mucosal samples were serially diluted and loaded onto the plates as well as serially diluted COVA1-18 as the standard. Following a 1 h RT incubation, goat anti-Human IgG (monkey adsorbed)-HRP secondary antibody (Southern Biotech) was added for serum samples (1:4000). For mucosal samples, goat anti-Human IgG (monkey adsorbed)-BIOT (Southern Biotech) was added at 1:10000 dilution. After 1 h RT incubation, serum sample plates were ready for development. For mucosal samples, an additional 1 h incubation with poly-HRP40 (Fitzgerald) (1:10000) was necessary. The plates were then developed for 5 min, and the optical densities measured at 450 nm on a spectrophotometer. The raw data were exported and analyzed using Microsoft Excel and GraphPad Prism (v8.3.0) software. The COVA1-18 concentration in a specific sample was determined by interpolating OD values from dilutions that fell into the linear range of the standard curve of the matching ELISA plate.

**Cynomolgus monkey IgG ELISA.** Half area high binding 96-well plates were coated overnight (4 °C) with goat anti-Human IgG λ and goat anti-Human IgG κ (Southern Biotech), 1:2000 (each) in PBS, 50 μl/well. The plates were washed (1X TBS – 0,05% Tween20) and blocked for 2 h at RT with 50 μl/well casein buffer. Serially diluted mucosal and serum samples were loaded onto the plates. Serially diluted polyclonal cynomolgus IgG (Molecular Innovations) was used as standard. Following a 1 h incubation at RT, mouse anti-Monkey IgG Fc-BIOT (Southern Biotech) was loaded onto the plate (1:50000). After 1 h at RT, poly-HRP40 was added (1:10000) and the plates incubated for 1 h. Finally, the plates were washed 5 times, developed for 5 min, and analyzed as described above.

Cynomolgus anti-S IgG and IgM ELISA were performed as described above except that $2 \mu g ml^{-1}$ SARS-CoV-2 S Fld His-tagged protein were coated in the sample wells instead of goat anti-Human IgG λ and goat anti-Human IgG κ. For the IgM ELISA, the standard was obtained from Molecular Innovations and the detection goat anti-Monkey IgM (μ-chain specific)-Biotin antibody from Sigma Aldrich and used at 1:20000 dilution. Strep-HRP (R&D systems) at 1:500 was used for IgG and poly-HRP40 for IgM.

**Pseudovirus neutralization assay.** Neutralization assays were performed using SARS-CoV-2 S-pseudotyped HIV-1 virus and HEK293T hACE2 cells as described previously[13]. In brief, pseudotyped virus was produced by co-transfecting expression plasmids of SARS-CoV-2$_{\Delta 19}$ S proteins (GenBank MT449663.1) with an HIV backbone expressing NanoLuc luciferase (pHIV-1$_{NL4-3}$ ΔEnv-NanoLuc) in HEK293T cells (ATCC, CRL-11268). After 2 days, the cell culture supernatants containing SARS-CoV-2 S-pseudotyped HIV-1 viruses were harvested and stored at −80 °C. HEK293T hACE2 cells were seeded 20,000 cells/well in a flat-bottom 96-well plates one day prior to the start of the neutralization assay. COVA1-18 IgG and His$_6$-tagged Fab as well as heat-inactivated serum samples were serially diluted in 3-fold steps using cell culture medium and then mixed with pseudotyped virus in a 1:1 ratio and incubated for 1 h at 37 °C. The mixtures were then added to the HEK293T hACE2 cells in a 1:1 medium to mixture ratio. The final starting concentration for IgGs was $20 \mu g ml^{-1}$ and $13.33 \mu g ml^{-1}$ for Fab. The cells were then incubated at 37 °C for 48 h followed by one PBS wash and lysis buffer addition. The luciferase activity in the cell lysates was measured using the Nano-Glo Luciferase Assay System (Promega) and GloMax Discover microplate reader. Relative luminescence units (RLU) were normalized to those from positive control wells where cells were infected with SARS-CoV-2 pseudovirus without IgG, Fab or serum. The inhibitory concentration ($IC_{50}$) and neutralization titers ($ID_{50}$) were determined as the IgG/Fab concentration or serum dilution at which infectivity was inhibited by 50%.

Pseudotyped Vesicular Stomatitis Virus (VSVΔG) particles displaying SARS-CoV-2$_{\Delta 19}$ S and containing a luciferase reporter were used as previously described[14]. Two-fold dilution series of COVA1-18 were prepared in complete medium, pseudotyped virus added and the mixture incubated for 1 h at 37 °C. The virus-antibody mixtures were then loaded onto plates seeded with Vero E6 cells 24 h prior this step. Following a 20 h incubation at 37 °C, the luciferase substrate was added to lysed cells and RLU determined and analyzed as described above.

**Ethics and biosafety statement.** All mice were housed in a temperature controlled environment (68–72 degrees Fahreheit, 50–60% humidity) with twelve hours of light per day at the Center for Comparative Medicine and Surgery (CCMS) at Icahn School of Medicine at Mount Sinai (New York, NY, USA). All experiments involving viral infections were carried out in a CDC/ USDA-approved BSL-3 facility at CCMS and animals were transferred into the facility four days prior to onset of experiments. Mice were housed in Allentown individually ventilated cages with ad libitum access to food and water. The mouse experimental study was approved by the Icahn School of Medicine at Mount Sinai Institutional Animal Care and Use Committee (IACUC-2017-0170 and IACUC-2017-0330).

Hamsters were housed in the ABSL-4 facility of the Galveston National Laboratory. The animal protocol # 2004049 was approved by the Institutional Animal Care and Use Committee (IACUC) of the University of Texas Medical Branch at Galveston (UTMB).

Cynomolgus macaques (Macaca fascicularis) originating from Mauritian AAALAC certified breeding centers were used in this study. All animals were housed in IDMIT infrastructure facilities (CEA, Fontenay-aux-roses), under BSL-2 and BSL-3 containment when necessary (Animal facility authorization #D92-032-02, Préfecture des Hauts de Seine, France) and in compliance with European Directive 2010/63/EU, the French regulations and the Standards for Human Care and Use of Laboratory Animals, of the Office for Laboratory Animal Welfare (OLAW, assurance number #A5826-01, US). The protocols were approved by the institutional ethical committee "Comité d'Ethique en Expérimentation Animale du Commissariat à l'Energie Atomique et aux Energies Alternatives" (CEtEA #44) under statement number A20-011. The study was authorized by the "Research, Innovation and Education Ministry" under registration number APAFIS#24434-2020030216532863. All information on the ethics committee is available at https://cache.media.enseignementsup-recherche.gouv.fr/file/utilisation_des_animaux_fins_scientifiques/22/1/comiteethiqueea17_juin2013_257221.pdf.

**Viruses and cells.** For the macaques studies, SARS-CoV-2 virus (hCoV-19/ France/ lDF0372/2020 strain) was isolated by the National Reference Center for

Respiratory Viruses (Institut Pasteur, Paris, France) as previously described[51] and produced by two passages on Vero E6 cells in DMEM (Dulbecco's Modified Eagles Medium) without FBS, supplemented with 1% P/S (penicillin at 10,000 U ml$^{-1}$ and streptomycin at 10,000 μg ml$^{-1}$) and 1 μg ml$^{-1}$ TPCK-trypsin at 37 °C in a humidified $CO_2$ incubator and titrated on Vero E6 cells. Whole genome sequencing was performed as described[51] with no modifications observed compared with the initial specimen and sequences were deposited after assembly on the GISAID EpiCoV platform under accession number ID EPI_ISL_406596. Sequencing analysis revealed two clonal mutations, one in the *S* gene (22661 G > T: V367F, non-synonymous) and one in the *ORF3a* gene (26144 G > T: G251V, non-synonymous), which were already present in the challenge inoculum.

**Animals and study design**. Seven week old female Balb/cJ mice (Jackson Laboratories Bar Harbor, ME) were anesthetized before being administered with $2.5 \times 10^8$ PFU of human adenovirus type 5 encoding the human angiotensin converting enzyme-2 receptor (Ad5-hACE2) 5-days prior to challenge with SARS-CoV-2, as previously described[29,30]. Animals were transferred to the BSL-3 facility where two groups of n = 5 mice per group received 10 mg kg$^{-1}$ of COVA1-18 intraperitoneally 24 h prior to, or post-infection with $10^4$ PFU SARS-CoV-2 in 50 μl PBS. A control group of n = 3 mice received 50 μl PBS. Mice were euthanized 3 d.p.i. and lungs harvested to quantify viral lung titers. Lungs were homogenized in PBS using a Beadblaster Microtube homogenizer (Benchmark Scientific). SARS-CoV-2 plaque assay was performed on 10-fold serial dilutions of lung homogenates prepared in 0.2% bovine serum albumin (BSA) in PBS that were plated onto a Vero E6 cells monolayer and incubated with shaking for 1 h. Inoculum was removed and plates were overlaid with Minimal Essential Media (MEM) containing 2% FBS/ 0.05% oxoid agar and incubated for 72 h at 37 °C. Plates were fixed with 4% formaldehyde overnight, stained with a mAb cocktail composed of SARS-CoV-2 spike and SARS-CoV-2 nucleoprotein (Center for Therapeutic Antibody Discovery; NP1C7C7) followed by anti-Mouse IgG-HRP (Abcam ab6823) and developed using KPL TrueBlue peroxidase substrate (Seracare; 5510-0030).

Female golden Syrian hamsters, aged 6–7 weeks, were randomly assigned to two groups of n = 5 and microchipped 24 h before SARS-CoV-2 challenge. On the day of challenge, hamsters were anesthetized with ketamine/xylazine and challenged by the intranasal route with $10^5$ PFU of SARS-CoV-2 diluted in sterile PBS in the total volume 100 μl. Body weight and body temperature were measured each day, starting at day 0. Twenty four hours post-challenge, hamsters were treated with 10 mg kg$^{-1}$ of COVA1-18 diluted in 0.5 ml of sterile PBS via the intraperitoneal route. The control group of animals received an equal volume of sterile PBS via the intraperitoneal route. All animals were euthanized 72 h post-infection with an overdose of anesthetic (isoflurane or ketamine/xylazine) followed by bilateral thoracotomy, and terminal blood and lungs were collected at necropsy. Right lungs were frozen in 5 ml L-15 Leibowitz medium (Gibco) with 10% FBS. Tissue sections were homogenized in bead beater tubes, weighed, and supernatants were titrated per standard protocol. Briefly, of 10-fold dilutions of supernatants at 100 μl per well were placed atop of Vero-E6 monolayers in 96-well plates, the plates were incubated for 1 h, supernatants were replaced by methyl cellulose overlay, incubated for 3 days at 5% $CO_2$ and 37 °C. The plates were fixed with formalin, removed from BSL-4 according the approved protocol, and plaques counted to determine the viral titers.

Ten female cynomolgus macaques aged 3–6 years were randomly assigned between the control and treated groups to evaluate the efficacy of COVA1-18 prophylaxis. The treated group (n = 5) received one bolus dose of COVA-18 human IgG1 monoclonal antibody (10 mg kg$^{-1}$) by the intravenous route in the saphenous vein one day prior to challenge, while control animals (n = 5) received no treatment. All animals were then exposed to a total dose of $10^6$ PFU of SARS-CoV-2 (BetaCoV/France/IDF/0372/2020; passaged twice in VeroE6 cells) via the combination of intranasal and intratracheal routes (day 0), using atropine (0.04 mg kg$^{-1}$) for pre-medication and ketamine (5 mg kg$^{-1}$) with medetomidine (0.05 mg kg$^{-1}$) for anesthesia. Animals were observed daily and clinical exams were performed at baseline, daily for one week, and then twice weekly, on anaesthetized animals using ketamine (5 mg kg$^{-1}$) and metedomidine (0.05 mg kg$^{-1}$). Body weight and rectal temperature were recorded and blood, as well as nasopharyngeal, tracheal and rectal swabs, were collected. Broncho-alveolar lavages (BAL) were performed using 50 ml sterile saline on 3 d.p.i. Chest CT was performed at 3 d.p.i. in anesthetized animals using tiletamine (4 mg kg$^{-1}$) and zolazepam (4 mg kg$^{-1}$). Blood cell counts, hemoglobin, and hematocrit, were determined from EDTA blood using a DHX800 analyzer (Beckman Coulter).

One male and one female cynomolgus macaques aged 3–6 years received the treatment as described above for the pharmacokinetic and pharmacodynamics (PK/PD) study. Blood was sampled before and 2, 4, 6, and 24 h post-treatment. Saliva, nasopharyngeal and tracheal fluids were sampled before and 24 h post-treatment. Twenty-four hours post-treatment, animals were euthanized and their lungs, heart, kidney, liver, spleen, trachea, and brain were sampled, rinsed with PBS and around 100 mg of tissue was homogenized in 500 μl of PBS with a Precellys and stored at −80 °C.

**Virus quantification in NHP samples**. Upper respiratory (nasopharyngeal and tracheal) and rectal specimens were collected with swabs (Viral Transport Medium, CDC, DSR-052-01). Tracheal swabs were performed by insertion of the swab above the tip of the epiglottis into the upper trachea at approximately 1.5 cm of the epiglottis. All specimens were stored between 2 °C and 8 °C until analysis by RT-qPCR with a plasmid standard concentration range containing an RdRp gene fragment including the RdRp-IP4 RT-PCR target sequence. SARS-CoV-2 E gene subgenomic mRNA (sgRNA) levels were assessed by RT-qPCR using primers and probes previously described[52,53] (Supplementary Table 4). The protocol describing the procedure for the detection of SARS-CoV-2 is available on the WHO website[54].

**Chest CT and image analysis**. Lung images were acquired using a computed tomography (CT) system (Vereos-Ingenuity, Philips) as previously described[25,26], andanalysed using INTELLISPACE PORTAL 8 software (Philips Healthcare). All images had the same window level of enuity, window width of 1,600. Lesions were defined as ground glass opacity, crazy-paving pattern, consolidation or pleural thickening as previously described[35,55]. Lesions and scoring were assessed in each lung lobe blindly and independently by two persons and the final results were established by consensus. Overall CT scores include the lesion type (scored from 0 to 3) and lesion volume (scored from 0 to 4) summed for each lobe as previously described[25,26].

**Viral sequencing**. 30 RNA samples from nasopharyngeal and tracheal swabs as well as BAL fluids at 3 d.p.i. were selected for sequencing along with the inoculum. cDNA and multiplex PCR reactions were prepared following the ARTIC SARS-CoV-2 sequencing protocol v2[56]. V3 primer scheme (https://github.com/artic-network/primer-schemes/tree/master/nCoV-2019/V3) was used to perform the multiplex PCR for SARS-CoV-2. All samples were run for 35 cycles in the two multiplex PCRs. Pooled and cleaned PCR reactions were quantified using QubitTM fluorometer (Invitrogen). The Ligation Sequencing kit (SQK-LSK109; Oxford Nanopore Technologies) was used to prepare the library following the manufacturer's protocol ("PCR tiling of COVID-19 virus", release F; Oxford Nanopore Technologies). Twenty-four samples were multiplexed using Native Barcoding Expansion 1–12 and Native Barcoding Expansion 13–24 kits (EXP-NBD104 and EXP-NBD114; Oxford Nanopore Technologies). Two libraries of 24 samples were prepared independently and quantified by QubitTM fluorometer (Invitrogen). After the quality control, two R9.4 flowcells (FLO-MIN106; Oxford Nanopore Technologies) were primed as described in the manufacturer's protocol and loaded with 45 and 32 ng of library. Sequencing was performed on a GridION (Oxford Nanopore Technologies) for 72 h with high-accuracy Guppy basecalling (v3.2.10). After sequencing, demultiplexing was performed using Guppy v4.0.14 with the option -require_barcodes_both_ends to ensure high quality demultiplexing. Reads were then filtered by Nanoplot v1.28.1 based on length and quality to select high quality reads. Then, reads were aligned on the SARS-CoV-2 reference genome NC_045512.2 using minimap2 v2.17. Primary alignments were filtered based on reads length alignment and reads identity. Reads were basecalled and demultiplexed with Guppy 4.0.14. The potential clonal and subclonal variants were detected with a custom pipeline based on ARTIC network workflow. Longshot v0.4.1 was used for variant detection. The potential subclonal variants were manually curated by comparing the generated VCF files and visual inspection of the alignments in IGV browser.

**Statistical analysis**. Statistical analysis of Syrian hamsters and hACE2 mice lung viral titers as well as for NHP gRNA and sgRNA were carried out using Mann-Whitney unpaired t-test in GraphPad Prism software (v8.3.0).

**Reporting summary**. Further information on research design is available in the Nature Research Reporting Summary linked to this article.

## Data availability
The viral sequencing data used in Supplementary Fig. 3 have been deposited in the SRA repository under the accession code PRJNA758764 (PRJNA758764 - SRA - NCBI (nih.gov)). All the other raw data generated in this study are provided in the Source Data file. Source data are provided with this paper.

## Code availability
The code used for the modeling part has been deposited in the Zenodo repository[57] under the https://doi.org/10.5281/zenodo.5140032.

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

## Acknowledgements

We thank Benoit Delache, Sébastien Langlois, Joanna Demilly, Emma Burban, Quentin Sconosciuti, Maxime Potier, Nina Dhooge, Pauline Le Calvez, Jean-Marie Robert, Tierry Prot and Christina Dodan for their help with the macaque experiments; Laetitia Bossevot, Marco Leonec, Laurine Moenne-Loccoz and Julie Morin for the RT-qPCR, and for the preparation of reagents; Blanche Fert and Céline Mayet for her help with the CT scans; Céline Aubenque, Karine Storck, Mylinda Barendji, Julien Dinh and Elodie Guyon for the macaque sample processing; Sylvie Keyser for the transports organization; Nastasia Dimant and Brice Targat for their help with the experimental studies in the context of COVID-19-induced constraints; Frédéric Ducancel and Yann Gorin for their help with the logistics and safety management; Isabelle Mangeot for her help with resources management. We thank Sylvie Behillil and Vincent Enouf for contribution to viral stock challenge production, Antoine Nougairede for sharing the plasmid used for the sgRNA assays standardization and Paul Bieniasz for donating cells and reagents for pseudovirus neutralization assays. We thank Matt Hyde and Julie Williams (Animal Resource Center, University of Texas Medical Branch at Galveston) who performed technical procedures with animals. We acknowledge support from CoVIC supported by the Bill and Melinda Gates Foundation. We thank staff at the ISMMS CCMS vivarium for their assistance. We also thank Randy Albrecht for support with the BSL-3 facility and procedures at the ISMMS and Richard Cadagan for excellent technical assistance. The macaque image used in Fig. 2 has been obtained from BioRender. This study was supported by the Netherlands Organization for Scientific Research (NWO) Vici grant (to R.W.S.), the Bill & Melinda Gates Foundation through the Collaboration for AIDS Vaccine Discovery (CAVD) grant INV-002022 (to R.W.S.), the Fondation Dormeur, Vaduz (to R.W.S. and to M.J.v.G.) and Health Holland PPS-allowance LSHM20040 (to M.J.v.G.). M.J.v.G. is a recipient of an AMC Fellowship, Amsterdam UMC and a COVID-19 grant of the Amsterdam Institute of Infection and Immunity, the Netherlands. R.W.S and M.J.v.G.

are recipients of support from the University of Amsterdam Proof of Concept fund (contract no 200421) as managed by Innovation Exchange Amsterdam (IXA). The Infectious Disease Models and Innovative Therapies (IDMIT) research infrastructure is supported by the "Programme Investissements d'Avenir", managed by the ANR under reference ANR-11-INBS-0008. The Fondation Bettencourt Schueller and the Region Ile-de-France contributed to the implementation of IDMIT's facilities and imaging technologies. The NHP study received financial support from REACTing, the Fondation pour la Recherche Médicale (FRM, France; AM-CoV-Path) and the European Infrastructure TRANSVAC2 (730964). The virus stock used in NHPs was obtained through the EVAg platform (https://www.european-virus-archive.com/), funded by H2020 (653316). Work performed at Duke University was supported by the CoVPN grant (NIH AI46705) (to D.C.M.). The Ad5-hACE2 mouse work was supported in part by NIAID R21AI157606 (L.C.), and was partially supported by CRIP (Center for Research for Influenza Pathogenesis), an NIAID supported Center of Excellence for Influenza Research and Surveillance (CEIRS, contract # HHSN272201400008C) (A.G.S.), by supplements to NIAID grant U19AI135972 and DoD grant W81XWH-20-1-0270, by the Defense Advanced Research Projects Agency (HR0011-19-2-0020), and by the generous support of the JPB Foundation, the Open Philanthropy Project (research grant 2020-215611 (5384) and anonymous donors to A.G.S. Part of this study was supported by the Bill and Melinda Gates Foundation through grants OPP1170236 and INV-004923 (I.A.W.) and through the Global Health Vaccine Accelerator Platforms (GH-VAP) and the Coronavirus Immunotherapy Consortium (CoVIC) (Nexelis). The funders had no role in study design, data collection, data analysis, data interpretation, or data reporting.

## Author contributions

P.M. and Y.A. conceived, designed, performed experiments, analyzed data, managed the project, and wrote the manuscript (original draft). A.M. conceived and developed the predictive model, and wrote the manuscript. R.M., N.D.B. performed, supervised, and analyzed macaque experiment. N.A.K. designed, performed, and analyzed the hamster experiment. M.S. designed, performed, and analyzed the mouse experiment. A.W.F. performed the mouse experiment. J.L.S. produced antibodies and performed ELISAs. A.G. contributed to the predictive model development. J.A.B., M.P., I.B., C.M., M.O.T., N.S.A., and L.G. performed neutralization assays. V.Ch., S.D., and A.I. performed sequencing, analyzed, and interpreted the data. A.J.R. analyzed the hamster histology data. S.J. and R.R. performed the mouse experiment. T.G.C., P.J.M.B., T.P.L.B., J.v.S., M.B., M.J.v.B., H.L., and M.Y. produced proteins. C.E.M. contributed to the hamster experiment. V.Co. contributed to performing and supervising macaque studies. T.N. and J.L. contributed to the macaque experiments and analysis. N.K. and F.R. contributed to the macaque experiment. C.C. and R.H.T.F. provided resources and supervision for the macaque studies contributed to the macaque experiments and analysis. D.C.M., I.A.W., G.J.d.B., and A.G.S. provided resources and funding. E.G. provided resources and supervision for the sequencing. L.C. conceived, designed, and performed the mouse study; acquired funding. A.B. designed and supervised the hamster study, provided funding. S.v.d.W. provided the virus for the macaque study; J.G. conceived and developed the predictive model, supervised, provided funding, and wrote the manuscript. M.J.v.G, R.L.G., and R.W.S. conceived, designed, supervised the project, acquired funding, provided resources, and wrote the manuscript. All authors contributed to the review and editing of the final manuscript.

## Competing interests

Amsterdam UMC filed a patent application on SARS-CoV-2 monoclonal antibody COVA1-18. The García-Sastre Laboratory has received research support from Pfizer, Senhwa Biosciences, 7Hills Pharma, Pharmamar, Blade Therapeutics, Avimex, Johnson & Johnson, Dynavax, Kenall Manufacturing, ImmunityBio, Merck, and Nanocomposix. Adolfo García-Sastre has consulting agreements for the following companies involving cash and/or stock: Vivaldi Biosciences, Contrafect, 7Hills Pharma, Avimex, Vaxalto, Pagoda, Accurius, Farmak, Pfizer, and Esperovax. The remaining authors declare no competing interests.

## Additional information

[1]Université Paris-Saclay, Inserm, CEA, Center for Immunology of Viral, Auto-immune, Hematological and Bacterial diseases (IMVA-HB/IDMIT), Fontenay-aux-Roses & Le Kremlin-Bicêtre, Paris, France. [2]Departments of Medical Microbiology of the Amsterdam UMC, University of Amsterdam, Amsterdam Institute for Infection and Immunity, Amsterdam, The Netherlands. [3]Université de Paris, INSERM, IAME, Paris, France. [4]Department of Pathology, University of Texas Medical Branch at Galveston, Galveston, TX, USA. [5]Galveston National Laboratory, Galveston, TX, USA. [6]Department of Microbiology, Icahn School of Medicine at Mount Sinai, New York, NY, USA. [7]Life and Soft, Le Plessis-Robinson, France. [8]Graduate School of Biomedical Sciences, Icahn School of Medicine at Mount Sinai, New York, NY, USA. [9]Department of Integrative Structural and Computational Biology, The Scripps Research Institute, La Jolla, CA, USA. [10]Department of Microbiology, University of Texas Medical Branch at Galveston, Galveston, TX, USA. [11]Duke Human Vaccine Institute & Department of Surgery, Durham, NC, USA. [12]Nexelis, Laval, Québec, Canada. [13]Internal Medicine of the Amsterdam UMC, University of Amsterdam, Amsterdam Institute for Infection and Immunity, Amsterdam, The Netherlands. [14]Department of Medicine, Division of Infectious Diseases, Icahn School of Medicine at Mount Sinai, New York, NY, USA. [15]The Tisch Cancer Institute, Icahn School of Medicine at Mount Sinai, New York, NY, USA. [16]Global Health and Emerging Pathogens Institute, Icahn School of Medicine at Mount Sinai, New York, NY, USA. [17]University of Maryland School of Medicine, Department of Microbiology and Immunology and Center for Vaccine Development and Global Health (CVD), Baltimore, MD, USA. [18]Molecular Genetics of RNA Viruses, Department of Virology, Institut Pasteur, CNRS UMR 3569, Université de Paris, Paris, France. [19]National Reference Center for Respiratory Viruses, Institut Pasteur, Paris, France. [20]Department of Microbiology and Immunology, Weill Medical College of Cornell University, New York, NY, USA. [21]These authors contributed equally: Pauline Maisonnasse, Yoann Aldon. ✉email: r.w.sanders@amsterdamumc.nl; m.j.vangils@amsterdamumc.nl; roger.le-grand@cea.fr

