## [Peer Review File · Nature Communications]

COVA1-18 neutralizing antibody protects against SARS-CoV-2 in three preclinical modelsREVIEWER COMMENTS

Reviewer #1 (Remarks to the Author):

Summary

The authors investigated the prophylactic and therapeutic potential of a highly potent anti-SARS-CoV-2 antibody (COVA1-18) in different animal models. In addition, the authors showed the biodistribution of the antibody in different organ systems and mathematically modeled and estimated COVA1-18 concentrations and how they correspond to treatment efficacy.

Overall, the manuscript is well written and the results presented are interesting and important. Although the true clinical potential of another Class 2 antibody that is sensitive to E484K-harboring circulating variants of concern is unclear, the presented data is informative for monoclonal antibody therapy in general.

Strength:

- The biggest strength of the manuscript is the thorough characterization of the mAb pharmacokinetic and biodistribution in different tissues, especially in the respiratory tract (Fig. 2 and Extended Data Fig. 3) but other organ systems as well. These are important and timely findings that can potentially inform dosing regimens and required tissue antibody levels as correlates of protection for other anti-SARS-CoV-2 mAb products that are currently in clinical development. Such granular data on antibody distribution can also be of importance for questions of potential viral sanctuary sites in the brain or viral antigen persistence in the gut (<https://www.ncbi.nlm.nih.gov/pmc/articles/PMC7255711/> and <https://www.nature.com/articles/s41586-021-03207-w>)

I suggest "promoting" some of the information in Extended Data Fig. 3 to Main Fig. 2.

Limitations:

- The extent of the viral sequencing analysis is not entirely clear. How many sequences were recovered? Have viral sequences been recovered beyond three days post viral challenge? This part of the manuscript needs more explanation and should be expanded upon. In the current form the statement "COVA1-18 did not select for S protein escape mutants when evaluated as PrEP in NHP" lacks substance.
- As discussed, COVA1-18 is a so-called Class 2 antibody that almost certainly will lose potency against circulating variants of concern that harbor the E484K mutation. Any experimental data that was generated since the submission of this manuscript should be included or cited.

Minor comments:

- Line 85: this should say *in vitro*
- Paragraph 90-100: I recommend stating more clearly and reminding the reader that the publication Brouwer et al. included neutralization activity measurements of COVA1-18 against authentic SARS-CoV-2
- Fig. 1d – it would be informative to use the same symbols that are being used in Extended Data Fig. 1e to visualize the importance of detectable serum neutralization on therapeutic activity and decrease in lung PFUs
- How do the authors explain the discrepancy between detectable COVA1-18 serum levels in all 5 macaques but the lack of neutralizing activity in 1 of the 5 animals?
- Line 330: Title of reference is corrupted.

Reviewer #2 (Remarks to the Author):

In their manuscript entitled "COVA1-18 neutralizing antibody protects against SARS-CoV-2 in three preclinical models", Maisonnasse et al. investigate the *in vivo* efficacy of a clinically-isolated, previously described (Brouwer et al., *Science*, 2020) neutralizing antibody directed against the

receptor binding domain on the S protein (COVA1-18) of SARS-CoV-2, which is active against both the dominant circulating variant (D614G) and the emerging B.1.1.7 "British" variant. Notably, the authors demonstrate that prophylactic COVA1-18 administration was distributed across numerous tissues, including lung, and reduced viral titers in three relevant pre-clinical models: hACE2 mice, Syrian hamsters, and cynomolgus macaques. In NHPs, i.v. COVA1-18 had a plasma half-life of 12.6 days with an estimated efficacy of over 99% in the nasopharynx and trachea 10 days following administration. Furthermore, mathematical modeling demonstrated that dosages as low as 1 mg/kg have the potential to prevent viral replication. These are all very important insights, the study was well conducted, well written, and the conclusions are supported by the presented data. I have some comments that would need to be addressed to further improve the quality of this important study.

Major Comments:

- The therapeutic protection from the development of lung lesions (Fig. 3c) is not particularly convincing, with the difference due to only 2 out of 5 controls having higher score than the Nab-treated animals and considering treated NHPs had BAL viral loads comparable with controls (Fig. 3a). Do the authors see reduction of lung infiltrating neutrophils or macrophages in treated animals? That piece of data will strength this specific conclusion.
- While the serum concentrations of COVA1-18 remain consistently high in NHPs (Extended Fig. 2a), the serum neutralization ID50 values are surprisingly highly variable (Extended Fig. 2b,d). For example, is there an explanation why the ID50 values in 4/5 treated NHPs have a minimum at d4 post-infection? Another concern is MF8, which exhibited no serum neutralization up to d7, despite having comparable pharmacokinetics (Extended Fig. 3a-c); however, this animal had among the lowest SARS-CoV-2 gRNA and sgRNA content (Extended Fig. 4a,c). The authors should discuss this variability.
- The authors should discuss more on the possible limitations for clinically using COVA1-18 as a SARS-COV-2 prophylactic. For example, although other antibody cocktails have been shown to prevent infection (Baum et al., Science, 2020), these therapies have only attained emergency authorization to be used in infected patients to attenuate disease progression. Possible limitations that would need to be addressed would include the route of administration (intravenous), and a determination if COVA1-18 can be repeatedly dosed and how often without the formation of anti-drug antibodies (ADA).

Minor Comments:

- Extended Fig 4 and Fig 3 contain the same data. Is there a way to combine the two figures for simplicity (i.e. overlay the individual data points with the averages)?
- In either the introduction or discussion, the authors may want to highlight specifically how the COVA1-18 NAb differs from those that are FDA-approved. Perhaps supplying a supplemental table comparing their characteristics (i.e. targeting domain, serum half-life, variants neutralized, reduction in infectivity, ect.) might be useful?

Mirko Paiardini

Reviewer #3 (Remarks to the Author):

Maisonnasse, Aldon and team investigate the potential of monoclonal antibody COVA1-18 against SARS-CoV-2. They use three accepted animal models, including non-human primates and evaluate prophylactic and therapeutic protection. Against a challenge virus that was isolated early in the pandemic the authors show pre and post infection protection. Given the potency of protection against a high-dose SARS-CoV-2 challenge the authors use models to determine the optimal doses to confer robust protection. This work is important but I have concerns that the antibody may not be as protective as described and a lack of experiential validation of the models.

1. In the most stringent model, hamster, efficacy was mixed with 2 of the 5 animals developing viral titers that were similar to untreated controls. Protection and reduction of viral replication in these animals should be further characterized with larger cohorts. It would be quite useful to see

example lung tissue.

2. As reported in Extended Data Figure 2, serum concentrations of COVA1-18 sharply increase upon infusion and then slowly decline with time. Neutralizing titers, however, fluctuate. A peak of neutralization rapidly declines by days 4-7 and then rebounds at day 10. Is this due to COVA1-18 or the animal's own immune response? For this reason neutralization assays on serum collected from untreated controls are absolutely required. Macaque antibody responses to SARS-CoV-2 (both IgG and IgM) need to be quantified. My expectation is that the humoral response would have developed at or after day 10. The data at these time points suggest as much.

3. The sharp drop-off in neutralizing titer at days 4-7 and the relative stability of serum COVA1-18 concentrations suggests that the antibody's protective potential is not as potent as described. Its prophylactic potential is likely overstated. In this model the pharmacokinetics appear to be poor. The predictive models are based on an assumption that all neutralizing and protective activity is due to COVA1-18. If supplemented by a humoral response the predictions of the model would be inaccurate.

4. The authors use their data to develop models to predict antibody doses needed to achieve robust protection. While useful, the power of any model is its predictive potential. The authors should demonstrate, at least in a small animal model, that their predictions are supported with experimental results.

Minor suggestions:

As it is established that avidity enhances antibody binding I am not sure of the comparisons between Fab and IgG and discussion of the findings is needed.

In the cited work by Shen et al., N501Y raises the IC₅₀/IC₈₀ >5 fold. This may be quite important in vivo.

In Figure 3 it would be helpful if the information for the individual animals and cohort averages were presented in the panes. The authors should also discuss that lung pathology in this model is not frequently observed.

RESPONSE TO REVIEWER'S COMMENTS

(original reviewer's comments in italics; our response in blue)

Reviewer #1

Overall, the manuscript is well written and the results presented are interesting and important. Although the true clinical potential of another Class 2 antibody that is sensitive to E484K-harboring circulating variants of concern is unclear, the presented data is informative for monoclonal antibody therapy in general.

We are happy that the reviewer appreciates the value of our study. We agree that future therapies will need to take into account emerging SARS-CoV-2 variants including ones with the E484K mutation. Such therapies will most likely involve combinations of antibodies to increase the spectrum of activity against viral variants. We have started to explore such combinations and included new data (see below).

Strength:

The biggest strength of the manuscript is the thorough characterization of the mAb pharmacokinetic and biodistribution in different tissues, especially in the respiratory tract (Fig. 2 and Extended Data Fig. 3) but other organ systems as well. These are important and timely findings that can potentially inform dosing regimens and required tissue antibody levels as correlates of protection for other anti-SARS-CoV-2 mAb products that are currently in clinical development. Such granular data on antibody distribution can also be of importance for questions of potential viral sanctuary sites in the brain or viral antigen persistence in the gut (<https://www.ncbi.nlm.nih.gov/pmc/articles/PMC7255711/> and <https://www.nature.com/articles/s41586-021-03207-w>). I suggest "promoting" some of the information in Extended Data Fig. 3 to Main Fig. 2.

We thank the reviewer for this encouraging comment. We have modified the related figures accordingly to include information on organ biodistribution to Fig. 2.

Limitations:

• The extent of the viral sequencing analysis is not entirely clear. How many sequences were recovered? Have viral sequences been recovered beyond three days post viral challenge? This part of the manuscript needs more explanation and should be expanded upon. In the current form the statement "COVA1-18 did not select for S protein escape mutants when evaluated as PrEP in NHP" lacks substance.

We regret the lack of clarity. We provide below the number of sequences recovered in each animal and type samples. We now also provide a new detailed table for each samples for the three mucosal compartments tested at 3 dpi (i.e. nasopharyngeal, tracheal, BAL) as **Supplementary Information**. We note that due to the remarkable efficacy of COVA1-18 treatment and low viral loads, the recovery of viral sequences was challenging and not possible at all beyond 3 days post infection (dpi) in treated animals. Moreover, high number of PCR cycles were required to detect viral RNA at early time points in treated animals due to extremely low viral levels in the samples. In order to provide more information on escape mutations, we now include new sequencing data on BAL and tracheal samples at 3 dpi and

include these data in the new Supplementary Fig. 3. These additional new results showed that a single S protein mutant was discovered in the BAL sample of the MF7 treated animal. We have amended the text to include the new data and improve clarity (lines 196-202).

- *As discussed, COVA1-18 is a so-called Class 2 antibody that almost certainly will lose potency against circulating variants of concern that harbor the E484K mutation. Any experimental data that was generated since the submission of this manuscript should be included or cited.*

This is an excellent point in light of the emergence of new SARS-CoV-2 strains including ones with the E484K mutation. We now tested COVA1-18 against the Wuhan, D614G, B.1.1.7 and B.1.351 strains and show that while COVA1-18 retains activity to the D614G and B.1.1.7 strains it does not neutralize the B.1.351 strain. As expected the E484K mutation was responsible for this loss. These new ELISA and pseudovirus data are now included in the manuscript (Supplementary Fig. 6a, b and Table 1). We also include new data on a cocktail of COVA1-18 with another mAb, COVA1-16. Although less potent than COVA1-18 against the Wuhan, D614G and the B1.1.7 strains, COVA1-16 remains active against the B.1.351 strain and can even cross-neutralize SARS-CoV-1. The additional data now provided in our revised manuscript using this cocktail demonstrate that the very high potency of COVA1-18 is maintained and that the addition of COVA1-16 effectively neutralizes the B.1.351 strain. We propose that a cocktail of the ultrapotent COVA1-18 with the broadly active COVA1-16 provides a path forward towards clinical application. The new data are included in Supplementary Fig. 6a, b and Table 2.

Minor comments:

- *Line 85: this should say in vitro*

This has been changed.

- *Paragraph 90-100: I recommend stating more clearly and reminding the reader that the publication Brouwer et al. included neutralization activity measurements of COVA1-18 against authentic SARS-CoV-2*

The text now mentions this.

- *Fig. 1d – it would be informative to use the same symbols that are being used in Extended Data Fig. 1e to visualize the importance of detectable serum neutralization on therapeutic activity and decrease in lung PFUs*

The symbols now match.

- *How do the authors explain the discrepancy between detectable COVA1-18 serum levels in all 5 macaques but the lack of neutralizing activity in 1 of the 5 animals?*

We have discussed this point in more detail in response to reviewer #2 below.

- *Line 330: Title of reference is corrupted.*

This has been corrected.

Reviewer #2

In their manuscript entitled “COVA1-18 neutralizing antibody protects against SARS-CoV-2 in three preclinical models”, Maisonnasse et al. investigate the in vivo efficacy of a clinically-isolated, previously described (Brouwer et al., Science, 2020) neutralizing antibody directed against the receptor binding domain on the S protein (COVA1-18) of SARS-CoV-2, which is active against both the dominant circulating variant (D614G) and the emerging B.1.1.7 “British” variant. Notably, the authors demonstrate that prophylactic COVA1-18 administration was distributed across numerous tissues, including lung, and reduced viral titers in three relevant pre-clinical models: hACE2 mice, Syrian hamsters, and cynomolgus macaques. In NHPs, i.v. COVA1-18 had a plasma half-life of 12.6 days with an estimated efficacy of over 99% in the nasopharynx and trachea 10 days following administration. Furthermore, mathematical modeling demonstrated that dosages as low as 1 mg/kg have the potential to prevent viral replication. These are all very important insights, the study was well conducted, well written, and the conclusions are supported by the presented data. I have some comments that would need to be addressed to further improve the quality of this important study.

We thank the reviewer for these positive comments.

Major Comments:

- *The therapeutic protection from the development of lung lesions (Fig. 3c) is not particularly convincing, with the difference due to only 2 out of 5 controls having higher score than the Nab-treated animals and considering treated NHPs had BAL viral loads comparable with controls (Fig. 3a). Do the authors see reduction of lung infiltrating neutrophils or macrophages in treated animals? That piece of data will strength this specific conclusion.*

The reviewer brings up an important point that merits careful attention. First, the NHP models mirrors what happens in humans in the sense that many animals do not develop severe COVID. Therefore, the heterogenous lung lesion scores in the control animals are expected. That is also the reason why we included the historic control data in addition to the data on the contemporaneous controls. We agree with the reviewer that this complicates gauging an effect of a treatment. We had a limited number of animals available for this study, precluding the inclusion of additional animals to be euthanised at early time points for the purpose of studying lung histopathology, in particular because we also wished to investigate the pharmacokinetics of COVA1-18 as well as the course of infection after COVA1-18 prophylaxis. As an alternative to lung histopathology, we therefore chose to use chest CT, a non-invasive method, to assess lung damage. We note that this approach is very similar to what is used in clinical practice in human COVID-19 patients.

- *While the serum concentrations of COVA1-18 remain consistently high in NHPs (Extended Fig. 2a), the serum neutralization ID50 values are surprisingly highly variable (Extended Fig. 2b,d). For example, is there an explanation why the ID50 values in 4/5 treated NHPs have a minimum at d4 post-infection? Another concern is MF8, which exhibited no serum neutralization up to d7, despite having comparable pharmacokinetics (Extended Fig. 3a-c); however, this animal had among the lowest SARS-CoV-2 gRNA and sgRNA content (Extended Fig. 4a,c). The authors should discuss this variability.*

This comment requires further clarification. While performing these assays, we determined that COVA1-18 in serum is sensitive to the heat-inactivation procedure required to test sera in neutralization assays. As a result, we measured limited neutralization in COVA1-18-treated animals. We determined that the concentration of COVA1-18 in heat-inactivated sera was ~28x lower (on average; range 14 to 50x) than in non-heat-inactivated sera. We show below the heat-inactivated serum COVA1-18 concentration and the total neat vs. heat-inactivated serum cynomolgus IgG concentrations. Given this finding and the confusion that inclusion of the data has caused, we propose to remove the neutralization data altogether.

• *The authors should discuss more on the possible limitations for clinically using COVA1-18 as a SARS-COV-2 prophylactic. For example, although other antibody cocktails have been shown to prevent infection (Baum et al., Science, 2020), these therapies have only attained emergency authorization to be used in infected patients to attenuate disease progression. Possible limitations that would need to be addressed would include the route of administration (intravenous), and a determination if COVA1-18 can be repeatedly dosed and how often without the formation of anti-drug antibodies (ADA).*

Indeed, our study does not address the clinical use of COVA1-18, directly. However, our pre-clinical data and modelling data support the investigation at the clinical level of this monoclonal antibody in a cocktail. In response to the reviewer's remark, we now include discussion along the lines the reviewer suggests (lines 289-292 and 301-306).

The route of administration is a valid point to address. While anti-SARS-CoV-2 mAbs tested in the clinic and approved for emergency use have been given intravenously, other therapeutic mAbs are given intramuscularly (palivizumab for RSV infection), or by subcutaneous injections (e.g. certolizumab pegol, adalimumab, efalizumab, and omalizumab). We also note that it might be possible to apply COVA1-18 and other SARS-CoV-2 mAbs intranasally which should result in strong presence where it is needed. An additional application could be a gene therapy that includes the gene for the mAb, in particular when such a gene therapy could target the lungs. Whether the biodistribution of COVA1-18 would be similar by different routes of administration should be investigated. We now include discussion on alternative delivery routes in lines 310-314.

Regarding ADA, we would like to point out that COVA1-18 and numerous potent neutralizing Abs isolated to date against SARS-CoV-2 present very low somatic hypermutation levels. Thus, these antibodies are very close to the germline precursor and unlikely to trigger anti-idiotypic response in patients. This is now also mentioned in the discussion (lines 314-317).

Minor Comments:

- *Extended Fig 4 and Fig 3 contain the same data. Is there a way to combine the two figures for simplicity (i.e. overlay the individual data points with the averages)?*

We have modified the figures accordingly.

- *In either the introduction or discussion, the authors may want to highlight specifically how the COVA1-18 NAb differs from those that are FDA-approved. Perhaps supplying a supplemental table comparing their characteristics (i.e. targeting domain, serum half-life, variants neutralized, reduction in infectivity, ect.) might be useful?*

We now include a supplemental table that compares SARS-CoV-2 mAbs from published NHP studies, including FDA-approved ones (Supplementary Table 3). We have also included the data (where available) on neutralization of the B.1.1.7 and B.1.351 variants, and included the same data on COVA1-16.

Reviewer #3

Maisonnasse, Aldon and team investigate the potential of monoclonal antibody COVA1-18 against SARS-CoV-2. They use three accepted animal models, including non-human primates and evaluate prophylactic and therapeutic protection. Against a challenge virus that was isolated early in the pandemic the authors show pre and post infection protection. Given the potency of protection against a high-dose SARS-CoV-2 challenge the authors use models to determine the optimal doses to confer robust protection. This work is important but I have concerns that the antibody may not be as protective as described and a lack of experiential validation of the models.

1. In the most stringent model, hamster, efficacy was mixed with 2 of the 5 animals developing viral titers that were similar to untreated controls. Protection and reduction of viral replication in these animals should be further characterized with larger cohorts. It would be quite useful to see example lung tissue.

The hamster data require further clarification. Indeed, the efficacy was mixed. However, while three animals showed a dramatic >1000-fold decrease in viral titer compared to controls, the viral titers in the remaining two animals was not similar to untreated controls, but decreased with a more modest 3.7 and 5-fold. Overall, despite the low number of animals (n=5), the difference between treated and untreated animals was highly statistically significant ($p=0.0079$). Considering this and the successful results in mice and NHPs, we think it is not ethically justified to repeat the experiment with more animals. Interestingly, the mixed efficacy was associated with neutralizing activity and we mentioned in the text that the two treated animals with the modest 3.7 and 5 fold reduction in viral titers also have the weakest neutralizing activity. As the injection was given intraperitoneally, it is possible that the distribution in these two animals was not optimal as reflected by the low systemic neutralization titers. In our opinion, the strong efficacy data COVA1-18 in the hACE2 mouse model, the NHP model and hamster model do not justify an additional hamster study.

We have quantified lung lesions by histopathology at day 3 as well as weight loss. We now provide these data in Supplementary Fig. 1f and g. It is well known that in this hamster model, SARS-CoV-2-related lung damage takes place within the first few hours after infection and the administration of mAbs at 24 hours post-infection is not expected to decrease lung damage and could only enable earlier recovery due to control of the virus. In other words, the damage is already done at the time of mAb administration and 3 days is too short to observe the effect on recovery. We also assessed weight loss and did find a modest but not statistically significant effect on weight loss. More clear data of weight loss/gain may be obtained within 14-days long observation but it would have doubled number of animals. These data are now included as Supplementary Fig. 1f and g and we now comment on these points lines 128-131.

2. As reported in Extended Data Figure 2, serum concentrations of COVA1-18 sharply increase upon infusion and then slowly decline with time. Neutralizing titers, however, fluctuate. A peak of neutralization rapidly declines by days 4-7 and then rebounds at day 10. Is this due to COVA1-18 or the animal's own immune response? For this reason neutralization assays on serum collected from untreated controls are absolutely required. Macaque antibody responses to SARS-CoV-2 (both IgG and IgM) need to be quantified. My expectation is that the humoral response would have developed at or after day 10. The data at these time points suggest as much.

The fluctuation in neutralization titers is due to heat-inactivation sensitivity as explained in more detail in our answer to reviewer #2. We considered the role of the animal's own immune responses in clearing the virus after the challenge. We have now determined the anti-S specific IgG levels in control and treated animals up to 6 and 28 days post-infection (dpi), respectively. We did not detect anti-S specific IgG in treatment or control groups throughout the course of the study (see new Supplementary Fig. 2h). In an effort to limit animal use in multiple SARS-CoV-2 studies at the time of the experiment, the control group was used for two studies and one required euthanasia of the control group at day 7 dpi which prevented us from evaluating the IgG response in this group beyond 7 days. We now evaluated the IgM response against SARS-CoV-2 S and showed that control animals developed IgM responses by 6 dpi while no IgM was detected in treated animals (see new Supplementary Fig. 2g). While the animal's own immune response may play a role in viral clearance – particularly innate immune responses at early time points – we think that the rapid control of viral replication in COVA1-18 treated animals is not consistent timewise, with a potential role for a *de novo* antibody response in clearing the infection, because the infection is cleared at 4 dpi while the IgM response does not become detectable before 6 dpi, as would be expected of an adaptive immune response. The analysis of T-cell responses at early timepoints was not possible due to the sampling schedule and number and volume of samples taken which prevented us from drawing more blood to isolate PBMCs. The new cynomolgus IgG and IgM data are now presented in Supplementary 2g and 2h and described in the results (lines 176-181).

3. The sharp drop-off in neutralizing titer at days 4-7 and the relative stability of serum COVA1-18 concentrations suggests that the antibody's protective potential is not as potent as described. Its prophylactic potential is likely overstated. In this model the pharmacokinetics appear to be poor.

We discussed the issues with measuring COVA1-18 neutralization activity in NHP sera related to heat-inactivation procedures in our reply to reviewer #2. The pharmacokinetics are largely comparable with those of other mAbs. We note that the decay rates of human mAbs tend to be faster in NHPs than in humans. For instance the half-life of bamlanivimab was 13 days in NHPs (Jones *et al.*, 2021, Science Translational Medicine, DOI 10.1126/scitranslmed.abf1906) but 18 days in humans (<https://www.fda.gov/media/143603/download>). This has now been clarified in the discussion “The plasma half-life was 12.6 days, albeit lower to what is found typically for human NAbs in humans, ranging from 15 to 25 days, is consistent with values reported for other human NAbs in the macaque model (Supplementary Table 3)” (lines 283-285).

The predictive models are based on an assumption that all neutralizing and protective activity is due to COVA1-18. If supplemented by a humoral response the predictions of the model would be inaccurate.

See our reply to question 2 for the relation between COVA1-18 activity and a *de novo* humoral immune response. The timing of the drop in viral loads is not consistent with a substantial effect of a *de novo* response. Furthermore, in a previous study we analysed the humoral response in a large cohort of animals and used the data for our models study (Gonçalves *et al.*, 2021, Plos Comp Bio, DOI 10.1371/journal.pcbi.1008785). None of the animals had detectable antibodies until day 7, and only 25% had detectable antibodies by day 14, suggesting that the humoral response played a minor role in viral clearance. Although it cannot be ruled out that other cellular or immunological mechanisms participate to the

antiviral efficacy, we think that these data suggest that the humoral response is unlikely to play a major role.

4. The authors use their data to develop models to predict antibody doses needed to achieve robust protection. While useful, the power of any model is its predictive potential. The authors should demonstrate, at least in a small animal model, that their predictions are supported with experimental results.

We thank the reviewer for this important comment, that gives us the possibility to clarify the point of these simulations. The main objective of our simulations was not to predict the outcome of experiments, but rather to guide and inform on the design of subsequent studies. We believe that in that respect, modelling can be extremely useful to adhere to the 3Rs principals and avoid useless or redundant experiments. We have clarified this aspect “Next, we used a viral dynamic model previously developed in the same SARS-CoV-2 NHP experimental model to evaluate the level of protection conferred by COVA1-18, and guide potential subsequent studies on SARS-CoV-2 MAbs” (lines 205-207).

Minor suggestions:

As it is established that avidity enhances antibody binding I am not sure of the comparisons between Fab and IgG and discussion of the findings is needed.

We agree with the reviewer that it is well known that antibodies frequently rely on avidity because of their bivalent binding, we thought it would be useful to include these data for two reasons. First, the diverse factors affecting neutralization potency such as affinity, avidity, hydrophobicity etc may vary between Abs. Second, we think it is useful to point out that avidity is important because it suggests that therapeutic strategies based on monovalent versions of COVA1-18 or other mAbs such as nanobodies might have less promise unless the avidity defect can be compensated for by other properties.

In the cited work by Shen et al., N501Y raises the IC50/IC80 >5 fold. This may be quite important in vivo.

We now added new data on COVA1-18 activity against the Wuhan, D614G and B.1.1.7 strains of which the latter contains the N501Y mutations (Supplementary Fig 6a, b and Table 1). We found only a marginal reduction (<2-fold) of the IC50 against the B.1.1.7 strain. As a result, COVA1-18 activity against B.1.1.7 remained very potent with an IC50 of 1.4 ng/mL.

In Figure 3 it would be helpful if the information for the individual animals and cohort averages were presented in the panes. The authors should also discuss that lung pathology in this model is not frequently observed.

We have now combined Fig. 3 and former Supplementary Fig. 4. The reviewer is right that lung pathology is not observed in each cynomolgus macaque infected by SARS-CoV-2, mirroring the heterogeneity of COVID-19 infection in humans. This is one reason why we also included comparative data on historic controls. We have now provided additional clarification on this in the text (lines 188-189). Lung CT lesions have been previously described in Maisonnasse *et al.* (Nature, 2020) and Brouwer *et al.* (Cell, 2021).

REVIEWERS' COMMENTS

Reviewer #1 (Remarks to the Author):

In this revised manuscript, the authors responded very carefully to the comments of the reviewers. They clarified most of the questions and provided new data regarding the neutralizing potency of COVA1-18 against variants of concern. This is a good paper and I have no further comments at this point.

Reviewer #2 (Remarks to the Author):

The authors provided a significant amount of new data and addressed in a satisfactory way my comments and, in my opinion, the main comments of the other reviewers.

Reviewer #3 (Remarks to the Author):

The revisions addressed all concerns that were raised. I hope this work informs continued development of monoclonal therapeutics.